# How can community pharmacists be supported to manage skin conditions? A multistage stakeholder research prioritisation exercise

Jane Harvey ![ORCID],[1] Zakia Shariff,[2] Claire Anderson,[3] Matthew J Boyd,[3] Matthew J Ridd ![ORCID],[4] Miriam Santer ![ORCID],[5] Kim Suzanne Thomas ![ORCID],[1] Ian Maidment ![ORCID],[2] Paul Leighton[1]

[1]Centre for Evidence Based Dermatology, School of Medicine, University of Nottingham, Nottingham, UK
[2]Aston Pharmacy School, Aston University School of Life and Health Sciences, Birmingham, UK
[3]Division of Pharmacy Practice and Policy, School of Pharmacy, University of Nottingham, Nottingham, UK
[4]Population Health Sciences, University of Bristol, Bristol, UK
[5]Primary Care Research Centre, University of Southampton, Southampton, UK

**Correspondence to**
Dr Jane Harvey;
jane.harvey1@nottingham.ac.uk

## ABSTRACT

**Objective** To establish research priorities which will support the development and delivery of community pharmacy initiatives for the management of skin conditions.

**Design** An iterative, multistage stakeholder consultation consisting of online survey, participant workshops and prioritisation meeting.

**Setting** All data collection took place online with participants completing a survey (delivered via the JISC Online Survey platform, between July 2021 and January 2022) and participating in online workshops and meetings (hosted on Microsoft Teams between April and July 2022).

**Participants** 174 community pharmacists and pharmacy staff completed the online survey.
53 participants participated in the exploratory workshops (19 community pharmacists, 4 non-pharmacist members of pharmacy staff and 30 members of the public). 4 healthcare professionals who were unable to attend a workshop participated in a one-to-one interview.
29 participants from the workshops took part in the prioritisation meeting (5 pharmacists/pharmacy staff, 1 other healthcare professional and 23 members of the public).

**Results** Five broad areas of potential research need were identified in the online survey: (1) identifying and diagnosing skin conditions; (2) skin conditions in skin of colour; (3) when to refer skin conditions; (4) disease-specific concerns and (5) product-specific concerns. These were explored and refined in the workshops to establish 10 potential areas for research, which will support pharmacists in managing skin conditions. These were ranked in the prioritisation meeting. Among those prioritised were topics which consider how pharmacists work with other healthcare professionals to identify and manage skin conditions.

**Conclusions** Survey responses and stakeholder workshops all recognised the potential for community pharmacists to play an active role in the management of common skin conditions. Future research may support this in the generation of resources for pharmacists, in encouraging public take-up of pharmacy services, and in evaluating the most effective provision for dealing with skin conditions.

## STRENGTHS AND LIMITATIONS OF THIS STUDY

⇒ Novel exploration of the research needs associated with the care of skin conditions within community pharmacy.
⇒ An iterative, multistage consultation ensured detailed insight about the topic.
⇒ The involvement of pharmacists, pharmacy staff, healthcare professionals and members of the public ensured that all pertinent voices were heard.
⇒ Participants were self-selecting and may have had a particular interest/perspective on skin conditions.
⇒ Greater participation from pharmacists in the prioritisation workshops may have been beneficial.

## INTRODUCTION

Community pharmacy is recognised as an accessible source of healthcare advice[1–3] and the COVID pandemic has cemented it more clearly in the primary care landscape for members of the public.[4] Moreover, recent initiatives, such as the Community Pharmacy Consultation Service (CPCS), seek to use pharmacy more effectively by diverting the management of some minor ailments to community pharmacy settings.[5]

Skin conditions are among the most common diseases encountered by healthcare professionals.[6 7] Each year, approximately 54% of the population will experience some form of skin disease,[6] at any one time up to one-third of all people will have a skin condition that warrants medical attention.[6 8 9] Skin reports have been identified as conditions that could be potentially managed within community pharmacy[10 11] and community pharmacists recognise skin conditions as a significant part of their workload.[1 12 13] Pharmacists regularly give advice on the management of common conditions such as eczema, dermatitis, generalised rashes, allergies and acne[12] and just over one-third (38%) of all

symptomatic advice requests in community pharmacy relate to skin conditions.[14] Almost 20% of pharmacy sales are for skin products.[2]

Due to the current stresses faced by the National Health Service, Community pharmacy in the UK is developing at a fast pace. Within England pharmacists are involved in treating dermatological conditions through the provision of a number of services. These include the CPCS, introduced in November 2020, where a General Practitioner (GP) surgery or National Health Service (NHS) 111 can refer patients to community pharmacies for the treatment of minor illness, for example, skin rashes.[15] Additionally, pharmacists may treat patients with skin conditions, free of charge, through minor ailments schemes, but this provision varies in availability between areas.[16] In some areas, specially trained pharmacists have access to prescription only medications through the use of patient group directions (PGDs) for certain conditions such as infected eczema or infected insect bites.[17] Most recently, the government announced that a 'Pharmacy First' scheme will be introduced within England. Through this scheme pharmacists will be able to prescribe medications (through PGDs) to treat conditions such as impetigo, shingles and infected insect bites.[18] In Wales and Scotland, the pharmacy first scheme has already been implemented. In these areas, medications can also be provided via PGDs or through independent (non-medical) prescribers.[19] For example, in Scotland, there are PGDs available for medications to treat impetigo, shingles and skin infections.[20]

Therefore, within the UK context pharmacists are already involved in the diagnosis and treatment of skin conditions and this involvement has accelerated in the past few years. As community pharmacy continues its trajectory towards expanded and extended provision[3] research will demonstrate the effectiveness of new ways of working and will support the development of new evidence-based services and resources.[6 8 21–23]

The aim of this work is to establish stakeholder consensus on those research priorities which might best support community pharmacists in their involvement in the care of patients with skin conditions.

## METHODS
This was a multistage, iterative stakeholder consultation informed by James Lind Priority Setting Partnership method[24] consisting of (1) an online survey, (2) exploratory workshops, and (3) a prioritisation workshop.

## Participants
### Stage 1—online survey
An online survey using the JISC Online Survey platform (https://www.jisc.ac.uk/online-surveys#) was targeted to community pharmacists and other community pharmacy staff. Social media (Twitter and Facebook) and personal and professional networks (eg, Pharmaceutical Services Negotiating Committee newsletter) were used to promote the survey. The survey was opportunistic and there were no specific inclusion criteria, that is, all pharmacists (and other members of pharmacy staff) were eligible to complete the survey.

A specific analysis of the survey data has been submitted for publication elsewhere.

### Stage 2—exploratory workshops
Community pharmacists, pharmacy staff, other healthcare professionals and members of the public were recruited to a series of workshops to explore potential research topics, which might support the management of skin conditions in community pharmacy. Equal numbers of public and professional participants were sought.

Social media (Twitter and Facebook) and personal and professional networks (eg, Community Pharmacy Dermatology Network, primary care networks) were again used to recruit pharmacists, pharmacy staff as well as healthcare professionals (eg, GPs, specialist nurse practitioners). Additional professional networks (the Primary Care Dermatology Society, the Society for Academic Primary Care Skin Special Interest Group and the UK Dermatology Clinical Trials Network) were used to recruit healthcare professionals.

Members of the public were recruited via social media and existing public and patient research networks (eg, the CEBD patient panel and 'People in Research' (https://www.peopleinresearch.org/)). All members of public who expressed an interest in the project were invited to join focus groups regardless of their experience of skin conditions or pharmacies.

Where any individual was not able to attend a scheduled workshop, they were offered the opportunity to take part in a brief one-to-one interview.

### Stage 3—prioritisation workshops
Exploratory workshop participants were subsequently invited to take part in the prioritisation workshop, with the goal of equal numbers of public and professional participants.

## Data collection and analysis
### Stage 1—online survey
Survey responses were collected over a period of 6 months between 20 July 2021 and 20 January 2022.

Content analysis of free text responses was used to identify commonly used words and phrases. Selected words or phrases (frequently used or substantively important) were reviewed thematically.[25]

### Stage 2—exploratory workshops and interviews
Workshops and interviews were undertaken online using Microsoft Teams and took place between April and July 2022. Up to six workshops were planned, to ensure 40–60 participants in this phase.

They were structured according to stage 1 data, with key themes explored further through group discussion. Discussion focused explicitly on 'research priorities'; although notions such as 'barriers', 'facilitators' and 'challenges' were also used to make discussions less

abstract and to support broad participation. See online supplemental files 1 and 2 for workshop schedules.

All discussions were digitally recorded with permission. Digital recordings were automatically transcribed verbatim and anonymised.

Framework analysis[26] was used to map workshop and interview data to broad uncertainties identified in stage 1. Synthesis of data and interpretation of synthesised data led to the creation of narrower research topics.

### Stage 3—prioritisation workshop

Following the conventions of the Nominal Group Technique,[27] research topics were shared with participants prior to the prioritisation workshop. During the workshop group discussion and item scoring were used iteratively to reject and rank topics. Simple, descriptive statistics were used to rank and establish consensus on priority research topics (ie, the percentage of respondents selecting a topic for inclusion/priority). For further information regarding methods, please see online supplemental file 3.

### Patient and public involvement

Before the study started, we met with two patient and public involvement (PPI) collaborators to provide an overview of the study and the study methodology. One PPI member collaborated with us to develop the participant information leaflet. They also attended the steering group meeting where we developed the final list of research questions. The other PPI collaborator assisted with recruitment of patients via social media and recommended other areas where we could recruit participants, for example, the 'people in research' website. They also attended one of our patient focus groups.

### RESULTS

The numbers and characteristics of participants that took place at each stage of the process are shown in table 1.

### Stage 1—online survey

The survey was completed by 174 participants. Word counts and an example of the word trees are available in online supplemental file 4.

The most reported five words in response to the 'challenge' questions were 'refer, rash, products, differential and know' while in the 'research priorities' question, the most reported words were 'treatment, different, products, need and creams'. These and the remaining top 20 words from each question encompass a broad range of research challenges, which were reflected in the five key areas of the analytic framework detailed in box 1.

### Stage 2—exploratory workshops and one to one interviews

Nine workshops were held, and four additional interviews to facilitate those unable to attend a scheduled workshop. Workshops lasted between one and 2 hours, interviews were typically around 30 min.

Four workshops consisted of pharmacists (19 participants), one included only pharmacy staff (4 participants),

---

**BOX 1 ORIGINAL ANALYTIC FRAMEWORK DEVELOPED FROM SURVEY RESPONSES**

⇒ Identifying and diagnosing skin conditions.
⇒ Skin of colour.
⇒ Knowing when to refer skin conditions to a GP.
⇒ Disease-specific concerns.
⇒ Product-specific concerns.

---

and four workshops contained only members of the public (30 participants). Interviews were undertaken with three GPs and one dermatology nurse specialist (table 1).

Data are presented here thematically, pointing to key uncertainties and research possibilities that these themes suggest (further examples of the data are available in online supplemental file 5).

### Theme 1—identifying and diagnosing skin conditions

The challenge of identifying and diagnosing skin conditions was a common focus and frequently described as a source of stress:

Skins a nightmare (workshop 2, pharmacist 1).

One of the worst things I can hear in a pharmacy is when a patient says, 'can I speak to the pharmacist? Can they tell me what this rash is my child has got? (workshop 2, pharmacist 2).

Difficulties identifying *reliable* resources were recognised. Google images, the NHS website, Clinical Knowledge Summaries website (CKS) or National Institute for Health and Care Excellent (NICE) guidelines were all discussed, but using *standard* photographs was not always found to be helpful. That some resources (eg, CKS and NICE) do not contain images further impacts on their utility.

Pharmacists explained that this is particularly an issue with skin conditions as members of the public commonly show them affected skin, rather than verbally describing symptoms as they do with other conditions.

---

**Table 1** Number of participants included at each stage of the priority setting exercise

| | Numbers of participants | | |
| --- | --- | --- | --- |
| | Stage 1 | Stage 2 | Stage 3 |
| Type of participant | | | |
| Patients | N/A | 30 | 23 |
| Pharmacists | 111 | 19 | 3 |
| Other members of pharmacy staff | 63 | 4 | 2 |
| Specialist dermatology nurse | N/A | 1 | 1 |
| GPs | N/A | 3 | 0 |
| GP, General Practitioner. | | | |

The development of pharmacy-specific resources (eg, online toolkits, in person training, etc) was recognised as a potentially important area for future research and action.

Possible research question—Would dedicated resources improve the identification of skin conditions in community pharmacy?

### Theme 2—Identifying and diagnosing skin conditions in skin of colour

Discussion of skin of colour proceeded almost as an extension of theme 1. With a few exceptions, most participants described identifying skin conditions in skin of colour as more difficult:

> I know fungal infection definitely look[s] different on like very dark skin, but I don't know whether my diagnosis would be right, so I'm just always doubting myself. Yeah, so one other thing is, um, discoid eczema (and) ringworm they look very similar on like dark skin or fair skin. So that comes up all the time. I get asked whether it's eczema ringworm all the time. And I don't know the difference (workshop 1, pharmacist 3).

Knowing when a condition was getting worse was also considered more challenging in skin of colour. Again, an absence of *reliable, high-quality, evidence-based* resources was considered a barrier to effectively responding to queries and questions.

Possible research question—Would dedicated resources improve the identification of skin conditions in skin of colour in community pharmacy?

### Theme 3—Knowing when to refer skin conditions

Members of the public described using community pharmacy as a form of triage, seeking advice about whether a condition was *'serious enough'* to consult other healthcare professionals. For some a pharmacist's advice had been an important factor in being confident enough to seek a doctor's appointment.

This was a role that pharmacists recognised but were not always comfortable with; they had specific concerns about *delaying diagnosis* of serious conditions, *missing infectious diseases* or a fear of *making a condition worse* by giving the wrong advice:

> Some condition can wait for next day or next week, but some condition need to be managed quite soon. Like same day referral. So I think my challenge was whether to refer […] because weekend 111 is so busy they take hours for them to the doctor they call them back. So sometimes they like go to walk in centre or wait. So I think it's either they can wait till next day or next few days or. With that same day, it's my challenge (workshop 1, pharmacist 3).

These concerns were considered more critical if advice was being sought about a child.

Pharmacists also identified that it was not always easy to contact other health professionals and that it is difficult to know when and how to refer patients. The potential for better connected services in the management of skin conditions was also recognised in one of our interviews (with a GP), although it was also recognised that resources might be a barrier to this.

Possible research question—Would dedicated resources support community pharmacists to effectively refer skin conditions that require urgent or more specialist attention?

### Theme 4—Disease-specific concerns

Workshop discussion did not confirm such a strong focus on specific skin conditions as the survey data, but rather pointed to general challenges of managing skin conditions in community pharmacy. An absence of feedback, and of knowing the outcome of advice was commonly described.

> So even though I've kind of recommended this steroid, or I've recommended this emollient, I don't know whether it worked or not because they just don't come back. Even if it's a regular customer (workshop 2, pharmacist 4).

This makes it harder for a pharmacist to feel fully confident in the advice that they are providing. Similarly, pharmacists rarely gained feedback when referring an individual to a GP, although subsequently seeing the GP's prescriptions might offer some informal insight.

The potential for pharmacists to be more involved in managing skin disease was commonly recognised in both the pharmacist workshops as well as healthcare professional interviews. A few suggested that this might be in diagnosing and suggesting initial treatments, others focused on counselling on long-term medication use:

> Perhaps a bigger and perhaps more important role for pharmacists is actually in supporting patients with long term chronic skin conditions. Because there are loads of people out there with eczema, acne, psoriasis and so on who don't really get the best out of their treatment and end up going into secondary care because they are very poorly managed …I think this is a golden opportunity for community pharmacists to get more involved is actually in supporting those patients (workshop 1, pharmacist 5).

Possible research question—How can community pharmacists work most effectively with other healthcare professionals in the identification and management of skin disease?

### Theme 5—Product-specific concerns

During the workshops pharmacists communicated that they were confident about their knowledge of products used to manage skin conditions. Members of the public reinforced the importance of this by communicating that

they expected pharmacists to understand the products that they were providing:

It seems to me that a pharmacist should be an expert on the products. And if they're not already an expert on the products then one questions what they're doing as a pharmacist. Sorry (workshop 5, patient 1).

During the pharmacist workshops, some frustration was communicated about not being allowed to provide certain products over the counter, products that customers would subsequently receive on prescription from their GPs:

I do recognize some or quite a few skin conditions, I would like to give them something that is prescription only, but I can't. So then I have to send them off to the GP, so I would personally like some sort of PGD [Patient Group Direction] or guidelines to be able to prescribe [erm], to do a course be accredited and to be able to prescribe that maybe not to have to go through the whole performance of becoming an independent prescriber, because I don't have the time or the facility to do that. (workshop 4, pharmacist 6).

Some members of the public were equally frustrated by this, confused about why they could order medications from the internet but not access them directly via community pharmacies. Topical corticosteroids (TCS) were often discussed in this way. Members of the public described how they had lied about how they were going to use TCS to ensure that it was provided:

the only time it was mentioned [topical corticosteroids] was when they refused to sell me it, you know, and that that that sounds stupid. It was, you know, when I've got it on prescription, there's never been any query or any conversation about it. It's just been given in a bag. [And] But when I needed to actually purchase something over the counter. And that's when the interrogation started (workshop 5, patient 2).

The centrality of product knowledge suggests that it could be an important area for research and resource development. The potential to extend what pharmacists can do may be important in this.

Possible research question—Could a wider range of products and treatments for skin conditions be made available via community pharmacy?

Possible research questions—Would dedicated resources support community pharmacists in the management of skin conditions?

### Theme 6—other topics

Discussion of the themes identified in the online survey often prompted a broader discussion of community pharmacy and skin conditions. This led to the identification of additional areas where research might be warranted.

In the workshops, pharmacists reinforced the notion that they see a broad range of skin conditions daily, and

that the number of customers seeking advice about skin conditions is increasing.

I think the numbers of skin referrals with the CPCS [Community Pharmacy Consultation Service] is going to go up into Community pharmacy because it's one thing that the GPs can triage without seeing (workshop 4, pharmacist 7).

Pharmacists, however, were less confident in making an assessment about how demand is growing and evolving, for example, which clinical conditions, what types of enquiry, specific demographic groups, adults/children, etc. Discussion in the workshops, as well as some one-to-one interviews, recognised that understanding trends in demand could be an important precursor to any substantive change in how community pharmacy works or engages with skin conditions.

Possible research question—In what ways are community pharmacists currently involved in the identification and management of skin conditions?

Some of these discussions (especially in the pharmacist workshop) exposed localised variation in what is available and what pharmacists are allowed to do.

We're lucky like, in England, as you have heard you saying that you guys have to charge your folk for it. In the pharmacy first, one thing we've got, is we can give it out free of charge and they can come back six times and get six different bottles and it doesn't cost them anything (workshop 2, pharmacist 1).

Discussion demonstrated that variations are manifest in: (1) the skin conditions that pharmacists are paid to treat via minor ailments schemes; (2) the medications which can be provided via the use of patient group directions (PGDs) and (3) whether patients had to pay for their treatment. As with understanding trends in demand it was considered pertinent to develop a better understanding of the success of current skin focused initiatives and ways of working as a precursor to any further development.

Possible research question—What are the known benefits of community pharmacy involvement in the identification and management of skin conditions?

The public workshops offered a slightly different perspective on this topic, focusing on establishing that pharmacists are appropriately qualified and competent to deal with skin conditions.

It just feels that pharmacists know a lot about their medicines and of course they know the pros and cons of uses, but whether they have got the expertise in recognizing a particular type of rash or a particular type of mark on the skin […] I wouldn't know if they had that expertise (workshop 7, patient 3).

Similar concerns were expressed in the healthcare professional interviews, with one GP indicating that they saw 'a lot of inappropriate' referrals from pharmacists.

Once again, the benefit of establishing the state of current provision for skin conditions in community pharmacy was recognised as an appropriate focus.

Possible research question—How competent are community pharmacists in the identification and management of skin conditions?

Discussion of direct experience of accessing community pharmacy exposed that participants in the public workshops were often polarised, between those that regularly used their pharmacist and those that were not aware that pharmacists offered this type of service.

> I never knew that they could advise you on skin you know problems, I never knew that. Because I thought they dealt dealt with drugs only and they all seemed very busy. So, I'd like to know how the pharmacist can help with skin conditions as well? (workshop 6, patient 4).

A concern about a lack of awareness about pharmacy services was echoed in the healthcare professional interviews, where GPs described difficulties convincing patients that minor ailments schemes are appropriate to use. Identifying barriers and encouraging the use of community pharmacy might be important research that underpins the effectiveness of any specific initiative:

Possible research question—What could be done to raise awareness of the skills that community pharmacists have with regards to the identification and management of skin conditions?

A list of all ten research questions is provided in box 2 (see online supplemental file 6) for research questions with explanatory notes).

### Stage 3—prioritisation workshops

To accommodate all those interested in participating, the prioritisation workshop was split into two parts, which ran consecutively. The first prioritisation workshops included one dermatology nurse specialist and 10 patients. The second prioritisation workshop included three pharmacists, two members of pharmacy staff and 13 patients (box 1). Scores at the end of the first workshop were carried forward as the starting point for the second.

At the conclusion of the second workshop the following questions was voted to be the most important (three participants did not vote):

► How can community pharmacists work most effectively with other healthcare professionals in the identification and management of skin disease? 11/15 participants.

► How competent are community pharmacists in the identification and management of skin conditions? 9/15 participants.

► Would dedicated resources improve the identification of skin conditions in community pharmacy? 6/15.

---

**BOX 2 THE 10 RESEARCH QUESTIONS FROM THE COMMUNITY PHARMACY AND DERMATOLOGY PRIORITY SETTING PARTNERSHIP (IN NO PARTICULAR ORDER, PRIORITISED QUESTIONS IN BOLD)**

⇒ **Would dedicated resources improve the identification of skin conditions in community pharmacy?**

⇒ Would dedicated resources improve the identification of skin conditions in skin of colour in community pharmacy?

⇒ Would dedicated resources support community pharmacists to effectively refer skin conditions that require urgent or more specialist attention?

⇒ **How can community pharmacists work most effectively with other healthcare professionals in the identification and management of skin disease?**

⇒ Could a wider range of products and treatments for skin conditions be made available via community pharmacy?

⇒ Would dedicated resources support community pharmacists in the management of skin conditions?

⇒ In what ways are community pharmacists currently involved in the identification and management of skin conditions?

⇒ What are the known benefits of community pharmacy involvement in the identification and management of skin conditions?

⇒ **How competent are community pharmacists in the identification and management of skin conditions?**

⇒ What could be done to raise awareness of the skills that community pharmacists have with regards to the identification and management of skin conditions?

---

## DISCUSSION
### Summary

Through consultation with a range of stakeholders including pharmacists, pharmacy staff, GPs, dermatology nurses and members of the public, we have developed a set of research questions to support dermatology provision in community pharmacy. In a final prioritisation exercise, we have established three topics as a starting point for a dermatology/community pharmacy research agenda.

Discussion in our workshops reinforced existing assessment that dermatology is a significant part of the workload faced by community pharmacists.[12 14 28] It also reinforced the expectation that this demand is likely to grow in future. Concerns about limited training and knowledge about skin conditions[1] were evident in comments about lacking confidence in dealing with skin queries. A perceived lack of resources, training and post-consultation feedback were recognised as factors in this. Developing resources for pharmacists/pharmacy staff might be an important area where research can benefit pharmacy practice—resources in this context might be training programmes, information resources for staff, information resources for the public, and they could be delivered in print, in person or online.

Research uncertainties identified here might suggest the value of repeating and/or expanding prior research, which has considered pharmacists' ability to identify skin conditions.[22 23]

Our findings also reinforce more general issues of public awareness about the role of community pharmacy,[29 30] specifically recognising this to be an issue with regard to skin conditions. Previous work has identified that individuals felt that only doctors are 'qualified/trustworthy' to manage skin complaints,[11] a view also expressed by medicine counter assistants.[1] This is a particularly important challenge to negotiate given that all stakeholders who took part in the exercise recognised that the demand in primary care exceeds what GPs are able to manage. However, as there is currently a shortfall in the number of pharmacists and pharmacy staff, it is possible that, under the current workforce model, pharmacists may also struggle to meet this demand.[31]

### Further information about the questions

We have developed a broad range of questions reflecting a broad range of concerns described in the workshops. It is important to note that though we have highlighted these questions for future research, we have not conducted a systematic review to check whether there is already research that addresses these issues.

We have framed questions loosely and in general terms to allow interpretation and wide scope for impact. For example, the final question, *'How can community pharmacists work most effectively with other healthcare professionals in the identification and management of skin disease?'* could be approached in terms of consideration of the appropriate prescribing of 'over the counter' medicines or in terms of the provision of information for regarding the use of long-term prescription medications.

In this, however, we might suggest that there is a form of natural hierarchy with questions focused on understanding current provision a necessary precursor to research which develops new ways of working. For example, answering, *in what ways are community pharmacists currently involved in the identification and management of skin conditions?*, would allow future work to be appropriately directed to the most common or most difficult to manage skin conditions.

We would encourage researchers to develop the focus of research in meaningful ways, mindful of the changing landscape within community pharmacy with initiatives such as the independent prescribing schemes in England.[3] We would also encourage researchers to consider how resources directed towards community pharmacists can deliver consistent messages to other healthcare professionals.

The term diagnosis in relation to pharmacist-led activity may not be as widely understand across the globe. However, our survey found identifying and diagnosing skin conditions, a key area for further research. This research could include the role of the pharmacist in diagnosing conditions compared with simply identifying them.

### Strengths and limitations

This has been a broad reaching exercise which has included 111 community pharmacists in an online survey as well as 57 workshop and interview participants. Workshops sought input from both healthcare professionals as well as members of the public – numbers were approximately even in the exploratory workshops (27 healthcare professionals / 30 members of the public).

While we are confident that a broad range of perspectives were considered in the identification of research topics, the final prioritisation stage was heavily weighted towards the input of members of the public. At this stage of our process, the goal of equal numbers was not achieved, with only five pharmacists/pharmacy staff participating in the prioritisation stage (despite 12 signing-up).

Overall, we were only able to recruit three GPs and one specialist dermatology nurse to take part in interviews. We did not include dermatologists or decision-makers in the project, reflecting our primary care focus. Further work in this area could explore different methods of engaging with GPs, other stakeholders and pharmacists to improve recruitment. We should consequently recognise that the final prioritisation more accurately reflects the public view of what research would be most beneficial, rather than a multiperspective assessment of this. We might also acknowledge that the qualitative nature of much of the data generated here necessarily required interpreting as part of data analysis—it may be that in our interpretations, we found questions and uncertainties that were not intended by participants. The iterative nature of the study with a final workshop specifically focused on research uncertainties hopefully tempers this process.

### CONCLUSION

Pharmacists are regularly consulted regarding skin conditions and do not always feel confident in the identification of skin disease. Using information from focus groups with pharmacists, members of the public and other stakeholders, we developed 10 research questions that can be used to direct future research to address these challenges.

**Acknowledgements** Thank you also to all the members of pharmacy staff, other healthcare professionals and members of the public who took part in interviews or workshops. Particular thanks goes to Amanda Roberts and our other patient representative for their help with this work.

**Contributors** ZS and IM conceptualised, designed and conducted the survey, ZS and JH analysed the survey data, JH and PL led the workshops and interviews, analysis of the interview and workshop data and drafted the manuscript, comments were made on the draft by ZS, IM, MS, MJR, MJB, KST and CA. All authors commented on and approved the final draft. PL is the guarantor for the work and/or conduct of the study, had access to the data and controlled the decision to publish.

**Funding** This project is funded by the National Institute for Health and Care Research (NIHR) School for Primary Care Research (project reference 522). The views expressed are those of the authors and not necessarily those of the NIHR or the Department of Health and Social Care.

**Competing interests** JH, IM, CA, KST, PL, ZS have declared they have no conflict of interest. MJB has received personal fees from Delphi Healthcare outside the submitted work, has grants or contracts with Health Education England (Grant for development of experimental learning activity) and Walgreen Boots Alliance (50% funding for PhD studentship). MJB has received consulting fees from Clinical Care Quality Solutions (payment split between institution and author). MJB has received

payment from the Ministry of Health Singapore, Human Manpower Development Programme (honoraria to speak including travel and accommodation). MJB is a Project Advisor/Chair of experiential learning during the MPharm research project. NHS Scotland (payment to institution and author). MJB is Vice Chair Pharmacy Law and Ethics Association (no fee received). MS has the following grants or contracts: RAPID and Efficient Eczema Trials (RAPID programme)—lead applicants Thomas and Roberts. NIHR PGfAR NIHR203279 funding to University of Southampton for 5% of my time.Trial of IGe tests for Eczema Relief (TIGER): randomised controlled trial of test-guided dietary advice for children with eczema, with internal pilot and nested economic and process evaluations – lead applicant Ridd. NIHR HTA NIHR133464 funding to University of Southampton for 10% of my time. Pragmatic, primary care, multi-centre, randomised superiority trial of four emollients in children with eczema, with internal pilot and nested qualitative study (Best Emollients for Eczema—BEE)—lead applicant Ridd. NIHR HTA 15/130/07 completed Aug 2020 funding to University of Southampton for 10% of my time. MS is a funding panel member NIHR Programme Grants for Applied Research 2018 to present day and also Academic PPIE lead and Board Member NIHR School for Primary Care Research 2022 to present day. MR has received various NIHR grants for studies of skin conditions/food allergy. MR is on TSC/DMC for ERICA, PRINCIPLE and ALPHA trials and is Co-Chair SAPC & NIHR SPCR skin/allergy research groups.

**Patient and public involvement** Patients and/or the public were involved in the design, or conduct, or reporting, or dissemination plans of this research. Refer to the Methods section for further details.

**Patient consent for publication** Not applicable.

**Ethics approval** This study was conducted in two parts. The survey was administered by Aston University and the workshops/interviews by the University of Nottingham. We included a copy of the Ethical Approval Letter from the Aston University Ethical Committee with regards to the survey in the submission of this article. Participants completing the survey were required to complete an online consent form.We also provided a copy of a letter from the University of Nottingham Faculty of Medicine and Health Sciences stating ethical approval was not required for this part of the study. Participants did not complete a written consent form in line with the fact this was not judged to be research, however, they were asked at the start of the focus groups/interviews whether they were happy to be recorded.

**Provenance and peer review** Not commissioned; externally peer reviewed.

**Data availability statement** Data are available upon reasonable request.

**ORCID iDs**
Jane Harvey http://orcid.org/0000-0003-1402-6116
Matthew J Ridd http://orcid.org/0000-0002-7954-8823
Miriam Santer http://orcid.org/0000-0001-7264-5260
Kim Suzanne Thomas http://orcid.org/0000-0001-7785-7465
Ian Maidment http://orcid.org/0000-0003-4152-9704

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
