## [Reviewer comments · BMJ Open]

ARTICLE DETAILS

TITLE (PROVISIONAL)	How can community pharmacists be supported to manage skin conditions? A multi-stage stakeholder research prioritisation exercise.
AUTHORS	Harvey, Jane; Shariff, Zakia; Anderson, Claire; Boyd, Matthew; Ridd, Matthew; Santer, Miriam; Thomas, Kim; Maidment, Ian; Leighton, Paul

VERSION 1 – REVIEW

REVIEWER	Howard, Matthew Victorian Melanoma Service
REVIEW RETURNED	12-Mar-2023

GENERAL COMMENTS	Interesting paper Fills a significant research gap with useful qualitative data I would be interested if any data on the following - pharmacists are often educated to advise people to apply topical steroids sparingly due to fear of adverse effects however many patient with significant atopic dermatitis etc require more liberal application
--

REVIEWER	Alsabbagh, Mhd Wasem University of Waterloo, School of Pharmacy
REVIEW RETURNED	16-Mar-2023

GENERAL COMMENTS	Thank you for giving me the opportunity to review this manuscript which aimed to determine research priorities pertaining to the development and delivery of an initiative by community pharmacists to care for patients with different skin conditions. The authors used multiple methods to gather this knowledge from relevant stakeholders such as community pharmacists, other health care professionals, other pharmacy staff members, and members of the general public. The techniques included online survey, focus groups and workshops, and meetings. They used online platform to replace in-person meetings when needed. Overall, the subject of this research is important and may be of interest to the readers of this journal. Identifying research priorities of pharmacy practice is essential to advance this practice to optimize patient care. I found this manuscript as a clearly written description a well performed research with appropriate methods and justifiable conclusions. However, at times, in the results section, I found that the issues elicited from participants more of practice issues pertaining to the facilitators and barriers impacting effective implementation rather than research priorities. I would suggest clarifying the language of
--

the themes that they are trying to get a picture of what are the most important issues facing pharmacists currently in this provision, which may trigger the need for research questions (where the third phase of the project prioritized).

I also made some specific notes to enhance the clarity of the manuscript.

Introduction

The introduction paints skillfully a picture of the importance of community pharmacist's role in primary care – especially after the pandemic – including minor ailment, and how skin conditions are common in this realm. However, the introduction does not justify the need for establishing the research priorities of the care of skin conditions by community pharmacists after the implementation of minor ailment management by pharmacists. Why – after the service is approved in several countries including the UK and Canada – do we need to establish the research priorities of skin conditions? To evaluate how these are applied and their impact. What is known about this subject? Such justification is essential to convince the reader of the importance of the research.

Methods

- I understand that the survey was submitted for publication but more details about the survey. Please provide briefly details of the PSNC and who were the pharmacists who were targeted with the survey. For example, minimum number of years of practice, practicing in the community with minimum number of hours.

- For the exploratory workshops: what did the “other health care professionals” mean? Please provide details of inclusion criteria – if any – of other health care professionals.

- Similarly, were there any inclusion criteria of members of the public to establish their close knowledge and relevant perspective of pharmacists care of skin conditions.

- Were there both quantitative and qualitative questions in the survey?

- Were there transcripts of the interview questions of the workshops?

- Was there a minimum sample size for each section of the study?

- Please define the (PPI collaborators).

- I did not have access to the Supplementary file 1 to see the additional details of the methods.

Results

- Did the authors exclude any participants who were not eligible?

- It seems that only GPs and one speciality nurse were recruited.

Were there any other primary care RNs, RPNs, NPs ..etc

- I applaud the participants and authors for the audacity using the word “diagnosis” while referring the assessment of skin conditions.

I do believe that this is essential for the furtherance of pharmacists' practice. In the jurisdiction where I practice and do research, people are told explicitly to avoid the “D” word in referring to minor ailment management by pharmacists or use the “assess” or “self-diagnosed” phrase. The justification was that pharmacists are not trained to be diagnosticians. I do suggest acknowledging that the use of “diagnose” might not be sanctioned in every jurisdiction.

- I suggest using skin conditions among “people of colour” rather than “skin conditions in skin of colour”.

Discussion

- I would suggest adding more context of what is currently being offered in the UK by pharmacists in the skin care in the community

	currently, and how the identified questions can be use to enhance this provision.  - Although the results in fact reinforce previous knowledge regarding issues like the workload, competency, and resources, I think the discussion can do more justice by highlighting the new knowledge this research yielded – even if the concepts are already known. For example, the resources of skin conditions among people of color is relatively an issue which was missed for a long time, at least where I work – where this limitation in textbooks and bulletins targeting pharmacists was identified recently. As such, a stronger language confirming the importance of this new knowledge would be justifiable. - I suggest disentangling the issue of general public awareness from general public trust in their community pharmacists. In earlier provisions – such as vaccination- although the first one was found a major barrier impeding larger implementation, the former was found as a non-issue (i.e., members of the public in fact trust their pharmacist).
--	---

REVIEWER	Choi, Ellie National University Hospital, Dermatology
REVIEW RETURNED	26-Apr-2023

GENERAL COMMENTS	This paper addresses an interesting topic on the role of community pharmacists in providing dermatologic advice/treatment. In the narrowest sense, this paper ranks potential topics for research, in a broader sense, it discusses some of the associated barriers and concerns from pharmacists and the public. Personally I felt that the premise for supporting pharmacists taking on a greater role in the care of patients with skin conditions needs to first be established. Should community pharmacists even be providing greater dermatological advice and treatment? Dermatological diagnoses are challenging to make compared to other diseases (even for GPs), as also evident by the pharmacists main concern re: misdiagnosis. So the current manuscript and discussion seems only relevant if we all agree that comment pharmacy should continue to expand and extend provision in the dermatologic setting. I'm also curious to know what was the rationale for this current study/interview of pharmacists?  - If it were to generate research ideas, then is it really necessary to conduct focused group interviews in this systematic way just to derive ideas? - It were to assess for gaps in the literature where further research may be warranted, then why not a scoping or systematic literature review instead? With the current approach, we have to be aware that research topics proposed by interviewees do not necessarily represent gaps in research (e.g. these questions may have already been investigated and published) Another concern is the relatively limited groups of stakeholders. There were only 3 doctors and 2 nurses out of 200+ participants. To compound the problem, there were no dermatologists nor policy planners/administrators which I personally feel would be important to include.
---

	I think it'll also be good to elaborate more on the methods (e.g. content, thematic and framework analysis) as it is not clearly described or presented. E.g. how did the authors come up with the list of themes and research questions. In terms of improving the manuscript flow, I personally feel there was heavy focus on each individual research question, many of which were fairly similar and overlapping. This made the results section in page 8-22 quite lengthy. In contrast, stage 1 and 3 were only very briefly mentioned in the results. I would have also liked to see more discussion on a broader range of topics for example the physician and patient perspective, evaluation of outcomes (whether this improves patient outcomes/healthcare utilisation/patient satisfaction etc), and any current implications for practicing pharmacists or dermatologists.
--	---

REVIEWER	Cowdell, Fiona Birmingham City University, Faculty of Health Education and Life Sciences
REVIEW RETURNED	02-May-2023

GENERAL COMMENTS	This is a well written manuscript about an important and topical subject. At present it is long and it would benefit from cutting the length / number of quotes to include only those most salient. Much is made of a dedicated resource for community pharmacists and while I understand this, I encourage you to think about how guidance for different professions and for citizens fits together so that we offer coherent and consistent information.
---

VERSION 1 – AUTHOR RESPONSE

Reviewer 1		
I would be interested if any data on the following - pharmacists are often educated to advise people to apply topical steroids sparingly due to fear of adverse effects however many patient with significant atopic dermatitis etc require more liberal application	Thank you, I'm afraid this was not a common theme that was discussed within the focus groups.	N/A
Reviewer 2		
I found that the issues elicited from participants more of practice issues pertaining to the facilitators and barriers impacting effective implementation rather than research priorities. I would suggest clarifying the language of the themes that they are trying to get a picture of what are the most important issues facing pharmacists currently in this provision, which may trigger the need for research questions (where the third phase f the project prioritized).	Thank you, we explored the barriers and facilitators within this project to inform the research questions where it was difficult to explain the concept of a research question. However, this was unclear, and we have clarified this in the aims within the methods.	“Workshops and interviews were undertaken online using Microsoft Teams and took place between April and July 2022. They were structured according to stage 1 data, with key themes explored further through group discussion. Discussion focused explicitly upon “research priorities”; although notions such as “barriers”, “facilitators” and “challenges” were also used to make discussions less abstract and to support broad participation.”
the introduction does not justify the need for establishing the research priorities of the care of skin conditions by community pharmacists after	Thank you, we have added an addition section where we clarify the current situation within the	We have added,

the implementation of minor ailment management by pharmacists. Why – after the service is approved in several countries including the UK and Canada – do we need to establish the research priorities of skin conditions? To evaluate how these are applied and their impact. What is known about this subject? Such justification is essential to convince the reader of the importance of the research	U.K context and the associated justification for the paper.	“Due to the current stresses faced by the National Health Service, Community pharmacy in the U.K. is developing at a fast pace. Within England pharmacists are involved in treating dermatological conditions through the provision of a number of services. These include the CPCS, introduced in November 2020, where a GP surgery or NHS 111 can refer patients to community pharmacies for the treatment of minor illness, for example skin rashes[15]. Additionally, pharmacists may treat patients with skin conditions, free of charge, through minor ailments schemes but this provision varies in availability between areas[16]. In some areas specially trained pharmacists have access to prescription only medications through the use of patient group directions (PGDs) for certain conditions such as infected eczema or infected insect bites[17]. Most recently the government announced that a “Pharmacy First” scheme will be introduced within England. Through this scheme pharmacists will be able to prescribe medications (through PGDs) to treat conditions such as impetigo, shingles and infected insect bites[18]. In Wales and Scotland, the pharmacy first scheme has already been implemented. In these areas medications can also be provided via PGDs or through independent (non-medical) prescribers[19]. For example, in Scotland there are PGDs available for medications to treat impetigo, shingles and skin infections[20]. Therefore, within the UK context pharmacists are already involved in the diagnosis and treatment of skin conditions and this involvement has accelerated in the past few years.”
---	--	---

I understand that the survey was submitted for publication but more details about the survey. Please provide briefly details of the PSNC and who were the pharmacists who were targeted with the survey. For example, minimum number of years of practice, practicing in the community with minimum number of hours.	Thank you for asking for clarification of the survey. The survey was opportunistic and there were no specific inclusion criteria i.e. all pharmacists (and other members of pharmacy staff) were eligible to complete the survey. Details of the survey have now been submitted for publication to the journal "Skin health and Disease". Within this publication there is more information regarding the demographics of survey participants. The PSNC, now known as community pharmacy England, represents community pharmacy contractors to the NHS.	We have added the following text and we will add the reference for the paper relating to the survey as soon as it is published. "The survey was opportunistic and there were no specific inclusion criteria i.e. all pharmacists (and other members of pharmacy staff) were eligible to complete the survey."
For the exploratory workshops: what did the "other health care professionals" mean? Please provide details of inclusion criteria – if any – of other health care professionals.	Thank you for requesting this clarification.	We have added: "other healthcare professionals (e.g. GPs, specialist nurse practitioners)"
Similarly, were there any inclusion criteria of members of the public to establish their close knowledge and relevant perspective of pharmacists care of skin conditions.	Thank you for requesting this clarification.	We have added "all members of public who expressed an interest in the project were invited to join focus groups regardless of their experience of skin conditions or pharmacies."
Were there both quantitative and qualitative questions in the survey?	Yes the survey included free text and categorical questions.	

Were there transcripts of the interview questions of the workshops?	Thank you we have added topic guides as an appendix.	
Was there a minimum sample size for each section of the study?	There is no formal sample size for this type of research, rather we sought broad engagement to ensure pertinent insight and discussion. We aimed to include 18-20 members of pharmacy staff and at least, 8-10 members of the public, 8-10 GPs in phase 2 and equal numbers of participants in phase 3 (approx. 10 public and 10 healthcare professionals). We did not manage to recruit these numbers of patients (particularly GPs in phase 2 and pharmacy staff in phase 3) and we acknowledge this in the limitations section of the paper.	
Please define the (PPI collaborators).	Thank you were have done this.	PPI (patient and public involvement)
I did not have access to the Supplementary file 1 to see the additional details of the methods.		For editorial team to sort please
Did the authors exclude any participants who were not eligible?	No we did not.	
It seems that only GPs and one speciality nurse were recruited. Were there any other primary care RNs, RPNs, NPs ..etc	No unfortunately we were unable to recruit any other practitioners.	
I applaud the participants and authors for the audacity using the word “diagnosis” while referring the assessment of skin conditions. I do believe	Thank you for raising this issue. We hope that by adding the text regarding the extent to which U.K. pharmacists are involved in the	We have added the following text to the discussion:

that this is essential for the furtherance of pharmacists' practice. In the jurisdiction where I practice and do research, people are told explicitly to avoid the "D" word in referring to minor ailment management by pharmacists or use the "assess" or "self-diagnosed" phrase. The justification was that pharmacists are not trained to be diagnosticians. I do suggest acknowledging that the use of "diagnose" might not be sanctioned in every jurisdiction.	treatment of skin conditions clarifies the fact that within the U.K. this is not at all controversial. However, we have also added additional text to the discussion section.	"The term diagnosis in relation to pharmacist-led activity may not be as widely understand across the globe. However, our survey found identifying and diagnosing skin conditions a key area for further research. This research could include the role of the pharmacist is diagnosing conditions compared to simply identifying them"
I suggest using skin conditions among "people of colour" rather than "skin conditions in skin of colour".	Thank you for this comment. However, we have used this terminology as per the British Association of Dermatologists website: https://www.bad.org.uk/education-training/skin-of-colour-in-dermatology-education/	
I would suggest adding more context of what is currently being offered in the UK by pharmacists in the skin care in the community currently, and how the identified questions can be use to enhance this provision.	As above	As above
Although the results in fact reinforce previous knowledge regarding issues like the workload, competency, and resources, I think the discussion can do more justice by highlighting the new knowledge this research yielded – even if the concepts are already known. For example, the resources of skin conditions among people of color is relatively an issue which was missed for a long time, at least where I work – where this limitation in textbooks and bulletins targeting pharmacists was identified recently. As such, a	Thank you for this comment. The 10 areas for future research we acknowledge are in some ways were unsurprising. However, in order for the drive towards the management of skin conditions within community pharmacy to be successful, it is crucial that all problematic areas are addressed and so we chose not to focus on just the new information.	

stronger language confirming the importance of this new knowledge would be justifiable.		
I suggest disentangling the issue of general public awareness from general public trust in their community pharmacists. In earlier provisions – such as vaccination- although the first one was found a major barrier impeding larger implementation, the former was found as a non-issue (i.e., members of the public in fact trust their pharmacist).	Thank you. In the focus groups the public consistently reported that they were not sure that pharmacists had the necessary skills to diagnose and manage skin conditions. This was because they did not know what training pharmacists had received and they were aware of complexities of diagnosing a skin condition. In contrast they also described that they trusted pharmacists to know about medicines as they understood this is what they had spent time studying. Therefore, we think it is still sensible to include confidence in pharmacists as an area for future research.	
Reviewer 3		
Personally I felt that the premise for supporting pharmacists taking on a greater role in the care of patients with skin conditions needs to first be established. Should community pharmacists even be providing greater dermatological advice and treatment? Dermatological diagnoses are challenging to make compared to other diseases (even for GPs), as also evident by the pharmacists main concern re: misdiagnosis. So the current manuscript and discussion seems only relevant if we all agree that comment pharmacy should continue to expand and extend provision in the dermatologic setting.	As above	As above

I'm also curious to know what was the rationale for this current study/interview of pharmacists?  - If it were to generate research ideas, then is it really necessary to conduct focused group interviews in this systematic way just to derive ideas? - It were to assess for gaps in the literature where further research may be warranted, then why not a scoping or systematic literature review instead? With the current approach, we have to be aware that research topics proposed by interviewees do not necessarily represent gaps in research (e.g. these questions may have already been investigated and published) 	Thank you. This study broadly followed the James Lind methodology for research prioritisation, which stresses the importance of different methods in research question generation. The JLA process also reminds us of the importance of all stakeholder voices being given an opportunity to inform the process. We recognise, indeed argue, that a systematic review of current initiatives is an important next step.	
Another concern is the relatively limited groups of stakeholders. There were only 3 doctors and 2 nurses out of 200+ participants. To compound the problem, there were no dermatologists nor policy planners/administrators which I personally feel would be important to include.	Thank you for raising this issue. Unfortunately, we had difficulties in recruiting GPs to the study. However, two members of the author team are GPs and we hope this would mean that we did not exclude the GP perspective. Although it would have been useful to speak to other stakeholders, this wasn't possible within the scope of the project. However, we still feel there is value in reporting the problems that pharmacists and pharmacy staff commonly encounter. We have included a recommendation in the discussion regarding	Added "other stakeholders" to the discussion.

	further work to include the views of these other stakeholders.	
I think it'll also be good to elaborate more on the methods (e.g. content, thematic and framework analysis) as it is not clearly described or presented. E.g. how did the authors come up with the list of themes and research questions.	We provided a supplementary file which explained how we used content analysis to develop themes from the survey responses. Were you able to see this file (other reviewers said they could not view supplementary information)?	
In terms of improving the manuscript flow, I personally feel there was heavy focus on each individual research question, many of which were fairly similar and overlapping. This made the results section in page 8-22 quite lengthy. In contrast, stage 1 and 3 were only very briefly mentioned in the results.	Thank you we have significantly edited the paper (please see track changed document)	
I would have also liked to see more discussion on a broader range of topics for example the physician and patient perspective, evaluation of outcomes (whether this improves patient outcomes/healthcare utilisation/patient satisfaction etc), and any current implications for practicing pharmacists or dermatologists.	Thanks-you for your comment. Such discussion would perhaps be interesting, but is in our opinion beyond the scope of our research prioritisation exercise.	
Reviewer 4		
At present it is long and it would benefit from cutting the length / number of quotes to include only those most salient.	Thank you we have significantly edited the paper (please see track changed document)	

Much is made of a dedicated resource for community pharmacists and while I understand this, I encourage you to think about how guidance for different professions and for citizens fits together so that we offer coherent and consistent information.	Thank you, we hope that this issue could be addressed by the question “How can community pharmacists work most effectively with other healthcare professionals in the identification and management of skin disease?” We have also added some text to address the concerns raised by your comment.	The following text has been added: “We would also encourage researchers to consider how resources directed towards community pharmacists can deliver consistent messages to other health care professionals.”
---	---	--

VERSION 2 – REVIEW

REVIEWER	Alsabbagh, Mhd Wasem University of Waterloo, School of Pharmacy
REVIEW RETURNED	11-Oct-2023

GENERAL COMMENTS	The authors have made significant improvements in addressing the comments from the previous review. The manuscript has been enhanced, but there are still two outstanding points that should be considered.  - The authors have provided valuable information regarding the involvement of pharmacists in the diagnosis and treatment of skin conditions in the UK, particularly highlighting the recent acceleration of this involvement. However, there remains a need for more robust justification of the need to establish research priorities. After the statement, "within the UK context, pharmacists are already involved in the diagnosis and treatment of skin conditions, and this involvement has accelerated in the past few years," it is crucial to emphasize the necessity for establishing research priorities. It would be beneficial to explain why it is essential to evaluate the success and impact of scaling up this service for patient care or similar justifications. This concern was also raised by Reviewer #3, who questioned the rationale for the study and recommended conducting a literature review to provide a stronger foundation for the research objectives. - The manuscript mentions the limitation of the low number of participants. To further enhance the transparency and completeness of the study, it is advisable to specify that the authors aimed to include a specific number of participants. For instance, you could state, "Our aim was to include 18-20 members of pharmacy staff, at least 8-10 members of the public, and 8-10 GPs in phase 2, with equal numbers of participants in phase 3 (approximately 10 from the public and 10 healthcare professionals)." Regardless of whether this number was determined through a formal sample size calculation, it is important to provide insight into why these specific numbers were sought. This will help readers understand the authors' approach and intentions in terms of participant selection. Overall, the manuscript has improved significantly, but addressing these two remaining points will further strengthen the research and its justification.
---

REVIEWER	Choi, Ellie National University Hospital, Dermatology
REVIEW RETURNED	19-Jun-2023

GENERAL COMMENTS	Many of the previous comments have been addressed and authors have expressed that these are limitations of the study. I do feel that they need to be better stated in the limitations section.  1. Lack of doctors, absence of dermatologists, decision makers 2. That research priorities may not be understood by participants, who instead reported on barriers and physical challenges. Challenges and difficulties highlight areas for correction/intervention, and that is quite different from areas for further research. For example, pharmacists highlighting difficulties
---

	with disease identification should not/does not equate to research question of “would dedicated resources improve the identification of skin conditions...” 3. Additionally as earlier mentioned, it is important to recognise that these proposed priorities do not necessarily represent true gaps in literature (e.g. there is probably already research evidence showing that education/training/resources does improve identification of skin disease) I was also unable to find the section on having obtained ethics approval and written patient consent (or otherwise approved by the ethics board).
--	--

VERSION 2 – AUTHOR RESPONSE

Reviewer 3		
I do feel that they need to be better stated in the limitations section.		
1. Lack of doctors, absence of dermatologists, decision makers	Thank you we have added a note to this effect.	Overall, we were only able to recruit 3 GPs and one specialist dermatology nurse to take part in interviews. We did not include dermatologists or decision makers in the project, reflecting our primary care focus.
2. That research priorities may not be understood by participants, who instead reported on barriers and physical challenges. Challenges and difficulties highlight areas for correction/intervention, and that is quite different from areas for further research. For example, pharmacists highlighting difficulties with disease identification should not/does not equate to research question of “would dedicated resources improve the identification of skin conditions...”	Thank you we have added a note to this effect.	We might also acknowledge that the qualitative nature of much of the data generated here necessarily required interpreting as part of data analysis – it may be that in our interpretations we found questions and uncertainties that were not intended by participants. The iterative nature of the study with a final workshop specifically focused upon research uncertainties hopefully tempers this process.
3. Additionally as earlier mentioned, it is important to recognise that these proposed priorities do not necessarily represent true gaps in literature (e.g. there is probably already research evidence showing that education/training/resources does improve identification of skin disease)	Thank you we have added a note to this effect.	It is important to note that though we have highlighted these questions for future research, we have not conducted a systematic review to check whether there is already research that addresses these issues.
I was also unable to find the section on having obtained ethics approval and written patient consent (or otherwise approved by the ethics board).	Thank you we have clarified the ethical process as described above.	AS ABOVE

Reviewer 2		
The authors have provided valuable information regarding the involvement of pharmacists in the diagnosis and treatment of skin conditions in the UK, particularly highlighting the recent acceleration of this involvement. However, there remains a need for more robust justification of the need to establish research priorities. After the statement, "within the UK context, pharmacists are already involved in the diagnosis and treatment of skin conditions, and this involvement has accelerated in the past few years," it is crucial to emphasize the necessity for establishing research priorities. It would be beneficial to explain why it is essential to evaluate the success and impact of scaling up this service for patient care or similar justifications. This concern was also raised by Reviewer #3, who questioned the rationale for the study and recommended conducting a literature review to provide a stronger foundation for the research objectives.	We have edited this part of the paper to reinforce the role that we feel research will play in the development and testing of new dermatology provision in community pharmacy.	As community pharmacy continues its trajectory towards expanded and extended provision[3] research will demonstrate the effectiveness of new ways of working and will support the development of new evidence-based services and resources.
The manuscript mentions the limitation of the low number of participants. To further enhance the transparency and completeness of the study, it is advisable to specify that the authors aimed to include a specific number of participants. For instance, you could state, "Our aim was to include 18-20 members of pharmacy staff, at least 8-10 members of the public, and 8-10 GPs in phase 2, with equal numbers of participants in phase 3 (approximately 10 from the public and 10 healthcare professionals)." Regardless of whether this number was determined through a formal sample size calculation, it is important to provide insight into why these specific numbers were sought. This will help readers	We have added comment to the research methods to emphasise our ambition of equal numbers of professional and public participants. We have indicated our pragmatic goal of up to 6 workshops to include 40-60 participants. this was a pragmatic decision which reflected the time and	Equal numbers of public and professional participants were sought. Exploratory workshop participants were subsequently invited to take part in the prioritisation workshop, with the goal of equal numbers of public and professional participants. Up to six workshops were planned, to ensure 40 – 60 participants in this phase.

understand the authors' approach and intentions in terms of participant selection.	resources available, and which might provide sufficient breadth of contribution and insight.	
--	--	--